# Influence of Environmental Temperature and Hormonal Stimulation on the In Vitro Sperm Maturation in Sterlet *Acipenser ruthenus* in Advance of the Spawning Season

**DOI:** 10.3390/ani11051417

**Published:** 2021-05-15

**Authors:** Viktoriya Dzyuba, Jacky Cosson, Maria Papadaki, Constantinos C. Mylonas, Christoph Steinbach, Marek Rodina, Vladimira Tučkova, Otomar Linhart, William L. Shelton, David Gela, Sergii Boryshpolets, Borys Dzyuba

**Affiliations:** 1South Bohemian Research Center of Aquaculture and Biodiversity of Hydrocenoses, Faculty of Fisheries and Protection of Waters, University of South Bohemia in České Budějovice, Zátiší 728/II, 389 25 Vodňany, Czech Republic; jacosson@gmail.com (J.C.); steinbach@frov.jcu.cz (C.S.); rodina@frov.jcu.cz (M.R.); vtuckova@frov.jcu.cz (V.T.); linhart@frov.jcu.cz (O.L.); wshelton@ou.edu (W.L.S.); gela@frov.jcu.cz (D.G.); sboryshpolets@frov.jcu.cz (S.B.); bdzyuba@frov.jcu.cz (B.D.); 2Hellenic Centre for Marine Research, Biotechnology and Aquaculture (IMBBC), Institute of Marine Biology, Heraklion, 71500 Crete, Greece; mpapadak@hcmr.gr (M.P.); mylonas@hcmr.gr (C.C.M.)

**Keywords:** sturgeon, kidney, Wolffian duct, seminal fluid, sperm maturation, spermatozoan motility, hormonal stimulation of spermiation, sex steroid hormones

## Abstract

**Simple Summary:**

Sperm maturation (acquisition of the potential for motility and fertilization by morphologically developed spermatozoa) in sturgeons is atypical of ray-finned fishes: it occurs outside the testes during the transit of testicular spermatozoa through the kidneys into the Wolffian ducts. We recently developed a method in which testicular spermatozoa of sterlet *Acipenser ruthenus* are matured in vitro when incubated in seminal fluid derived from Wolffian duct sperm. In this study, we explored whether in vitro maturation of testicular spermatozoa depends on the environmental temperature and/or hormonal stimulation of spermiation. We studied spermatozoa motility parameters after in vitro maturation of testicular sperm, concentrations of sex steroid hormones and testis morphology in sterlet males at different stages of male preparation for spawning with and without hormonal induction of spermiation. The obtained data suggest that the ability of testicular spermatozoa to be matured was not related to the environmental temperature, while hormonal stimulation was an absolute requirement for optimal in vitro maturation. The use of in vitro matured testicular spermatozoa might have considerable potential in aquaculture or conservation programs, which can be realized in cases of accidental death of valuable broodstock or failure to obtain Wolffian duct sperm of high quality.

**Abstract:**

Sturgeon sperm maturation occurs outside the testes during the transit of testicular spermatozoa (TS) through the kidneys and the Wolffian ducts. A method of in vitro TS maturation in sterlet *Acipenser ruthenus* was used to investigate the effects of temperature and hormonal stimulation of spermiation on the ability of TS to complete this process. Spermatozoa motility parameters after in vitro maturation of testicular sperm, concentrations of sex steroid hormones and testis morphology were studied in three groups of sterlet: (1) after overwintering in ponds (OW), (2) adapted to spawning temperature (ST), and (3) adapted to spawning temperature with hormonal induction of spermiation (ST-HI). Blood plasma concentrations of testosterone, 11-ketotestosterone and 17,20β-dihydroxy-pregnenolone increased significantly after hormonal induction of spermiation (group ST-HI). In all groups, TS were not motile. After in vitro sperm maturation, motility was up to 60% only in group ST-HI. The data suggest that the ability of TS to be matured in vitro was not related to the environmental temperature, while hormonal stimulation of spermiation during the spawning season was an absolute requirement for optimal in vitro maturation.

## 1. Introduction

Sturgeons (Acipenseridae) is an extant early diverged actinopterygian fish family [1]. The morphology of their reproductive system is quite different from that in teleostean species. The sperm of teleosts passes directly from the testes to the environment. Thus sperm consists of spermatozoa and seminal fluid, while testicular sperm of sturgeons passes through the kidneys and the Wolffian ducts, where it is mixed with urine. Thus sturgeon sperm contains spermatozoa, seminal fluid and urine (for details, see review [2]). This anatomical feature determines a unique sperm maturation process: testicular spermatozoa (TS) acquire the ability for motility activation and fertilization only after mixing with urine [3,4]. Recently, we have shown that final spermatozoan maturation in sturgeons can be achieved within in vitro conditions by pre-incubation of testicular sperm in urine or in the seminal fluid of sperm collected from Wolffian ducts which is the site where sperm and urine are mixed naturally [3].

It is important to note that the main aspects of sturgeon reproduction are quite well studied because of the considerable interest in sturgeon conservation and the great demand for caviar production [5]. It is clear that to obtain sturgeon sperm of high quality for artificial propagation, hormonal stimulation of spermiation should be used. This is why gonadotropins (e.g., sturgeon or carp pituitary extracts), which stimulate the synthesis of sex steroid hormones (androgens, estrogens, and progestogens) in gonads, are used in sturgeon fisheries practice [6]. This hormonal treatment is performed, considering the corresponding requisite temperature regime of conditioning sexually mature fish before hormonal treatment [7,8]. Data related to the effect of hormonal stimulation on sturgeon sperm production and quality were obtained using sperm collected from Wolffian ducts, where natural maturation of sperm occurs after mixing with urine. This is why it is not clear from these studies at which stage of spermatogenesis testicular spermatozoa acquire the ability for final maturation. Two scenarios are possible: (1) testicular spermatozoa are already prepared for final maturation, and that hormonal stimulation only provokes the production of seminal fluid that transports spermatozoa from the testis via the kidney to the Wolffian ducts, or (2) testicular spermatozoa are not ready for final maturation, and that gonadotropic hormones stimulate the last stage of spermiogenesis with the production of seminal fluid, but also produce specific intratesticular conditions, which prepare testicular spermatozoa for further final maturation upon contact with urine.

Identifying which scenario occurs naturally is essential for both a basic understanding of sperm maturation processes and the possible application of testicular sperm in sturgeon breeding. Thus, this study aimed to evaluate the ability of sterlet TS to complete final maturation at different stages of the male reproductive cycle via analysis of testicular spermatozoa motility parameters after in vitro maturation, concentrations of sex steroid hormones and testis morphology.

## 2. Materials and Methods

### 2.1. Ethics Statement

All manipulations with animals were performed following the authorization for using experimental animals (reference number: 2293/2015-MZE-17214) and the authorization for breeding and delivery of experimental animals (reference number: 56665/2016-MZE-17214) issued to the Faculty of Fisheries and Protection of Waters, University of South Bohemia, by the Ministry of Agriculture of the Czech Republic.

### 2.2. Fish Rearing Conditions and Experimental Groups

Experiments were performed using mature male sterlet *Acipenser ruthenus* (1.83 ± 1.13 kg; age 5–6 years), which were maintained at the Genetic Fisheries Center at the Faculty of Fisheries and Protection of Waters, Vodnany, Czech Republic. Fish were subdivided into three groups: (1) after overwintering in ponds, but without experimental temperature acclimation (OW); (2) after overwintering then adapted to spawning temperature 14 °C (ST) and (3) adapted to spawning temperature after overwintering and followed by hormonal induction of spermiation (ST-HI).

Fish from group OW (*n* = 7) were held in fish-farming ponds (water temperature 2 °C; fish were not fed, pond water pH was 8.3–8.7, contents of NH_4_^+^ and NO_3_^−^ were around 0 mg/mL, the oxygen content was higher than 95% of saturation level). Fish from group ST (*n* = 9) were transferred from fish-farming ponds (water temperature 2 °C) into the 0.8 m^3^ closed water recirculation system located at the hatchery. To adapt fish to spawning temperature, the water temperature was increased to 14 °C within six days, and fish were held at this temperature for the next six days. Following that, sperm was collected without hormonal treatment of males. Fish from group ST-HI (*n* = 11) were adapted to spawning temperature as it was done for fish from group ST, then males were injected with carp pituitary powder (CPP) dissolved in 0.9% (*w/v*) NaCl solution (4 mg/kg bw) for sperm collection after 24 h. Temperature regime and hormonal injection timing for experimental fish groups are presented in Figure 1.

### 2.3. Collection of Sperm, Blood and Testis Samples

Two types of sperm samples were collected: (1) Wolffian duct sperm (WS; mature, used in fisheries for artificial sturgeon propagation); it was collected from the urogenital (Wolffian) ducts by aspiration using a plastic catheter (4 mm diameter) connected to a 10 mL syringe, 24 h after stimulation of spermiation (group ST-HI), and (2) testicular sperm (TS; immature); it was collected from groups OW, ST, ST-HI after the fish were euthanized by striking the cranium followed by exsanguination. After euthanasia, the digestive tract was removed, and TS was collected after incision of the efferent ducts [9]. In group ST-HI, TS was collected immediately after WS collection.

Blood was collected from the caudal vein using a heparinized syringe before sperm collection. Blood samples were centrifuged at 5000× *g* for 10 min to separate plasma. The samples of separated plasma were stored at −80 °C for future assays of sex steroid hormones.

After fish dissection and collection of TS, pieces of testis were incised (from approximately the same area in the middle part of the organ) for evaluation of sex steroid hormones and histological analysis. To measure the sex steroid hormone concentrations, testis pieces were frozen at −80 °C and stored until analysis. For histological analysis, tissue samples were fixed in Bouin’s solution for 48 h.

### 2.4. Histological Analysis

The fixed samples were paraffin-embedded and routinely processed for histological examination. Serial cuts with a thickness of 4 μm were prepared, stained with hematoxylin–eosin, and examined through light microscopy. The determination of testicular development was based on identifying germ cells at different stages of the spermatogenesis following the definitions of Amiri et al. 1996 [10] as spermatogonial proliferation (stage 1), early spermatogenesis (stage 2), mid-spermatogenesis (stage 3), late spermatogenesis (stage 4), pre-spermiation (stage 5), and degeneration stage (stage 6).

### 2.5. Plasma Sex Steroid Hormone Evaluation

The enzyme-linked immunoassays (ELISA) used for the quantification of testosterone (T), 11-ketotestosterone (11-KT) and 17,20β-dihydroxypren-4-en-3-one (17,20β-P) in the plasma and gonads of fish were performed according to [11,12,13], respectively, with some modifications and using reagents from Cayman Chemical Company (Ann Arbor, MI, USA). For steroid extraction, 200 μL of plasma were extracted twice with 2 mL of diethyl ether. Extraction was done by vigorous vortexing (Vibramax 110, Heidolph, Germany) for 3 min. After decanting the organic phase (supernatant), drying twas done under a stream of nitrogen (Reacti-vap III, Pierce, Germany), and the samples were reconstituted in 250 μL of reaction buffer for running in the ELISA.

Before steroid extraction, testes were homogenized in 1 mL NaOH (0.5 M) with an electric tissue homogenizer (Yellow line DI25, IKA-Works, Germany) and sonicated for 30 s, at 30% amplitude (UP200S, Dr. Hielscher GmbH, Teltow, Germany). Steroid extraction was performed twice with 8 mL of diethyl ether. The procedure that followed was as described above for plasma. Samples were finally reconstituted in 500 μL of reaction buffer. The results were presented as ng per mL of plasma or g of tissue.

### 2.6. In Vitro Maturation of Testicular Spermatozoa

The recently developed method of in vitro TS maturation [3] involves TS incubation in urine or in seminal fluid derived from mature Wolffian duct sperm. In the present study, seminal fluid from WS was used for in vitro maturation (IVM). For IVM experiments, WS of sufficient volume (11.3 ± 1.3 mL) and high motility percentage (more than 70%) was available from group ST-HI only. The WS from groups OW and ST was usually of low volume (1.9 ± 0.3 and 2.0 ± 0.2 mL, respectively), and motility percentages were less than 10% (motility percentage in WS samples was estimated within half an hour after sperm collection). Thus, the seminal fluids from Wolffian duct sperm from only group ST-HI were used for IVM. To obtain seminal fluid from WS devoid of spermatozoa, WS samples were centrifuged at 1000× *g* at 4 °C for 10 min followed by 15 min at 5000× *g*. Supernatants were collected and centrifuged at 10,000× *g* at 4 °C for 15 min. Supernatants obtained after the last centrifugation were used for IVM of TS. For maturation, TS was incubated in seminal fluid from WS at a dilution rate of 1 volume of TS to 50 volume of seminal fluid for 30 min. Incubation was done at room temperature. After checking the percentage of sperm motility in TS from three males after incubation in three different seminal fluids from Wolffian duct sperm of the highest volume, seminal fluid from Wolffian duct sperm of only one male was selected for future experiments. The decision to select this seminal fluid was made based on the absence of differences in post-maturation motility percentage in samples incubated in different seminal fluids from the Wolffian duct.

### 2.7. Sperm Motility Analysis

Motility parameters of sperm samples from Wolffian duct sperm and testicular sperm after in vitro maturation were evaluated after sperm motility initiation by dilution 0.2–1 µL of sperm in 50 µL of activating solution consisting of 10 mM Tris-HCl, pH 8.0, and 0.25% Pluronic F-127, and video-recorded for 120 s using a negative phase-contrast microscope (UB 200i, 10× lens, PROISER, Spain) with an attached ISAS 782 M digital camera (PROISER, Valencia, Spain) set at 25 FPS. Sperm dilution was made on the surface of the microscopy slide, and microscope focusing was made on the bottom of the drop. The composition of activating solution and dilution rate were selected based on our previously published results [3].

The video records were analyzed using ImageJ software with CASA plugin [14] modified according to Purchase and Earle, 2012 [15]. The CASA data were obtained for each second of motility (starting from 10 s post activation of motility). The results of CASA analysis were initially accumulated into data set consisting of the following parameters: percent of motile cells, curvilinear velocity (VCL, µm/s), average path velocity (VAP, µm/s), straight-line velocity (VSL, µm/s), linearity of the track (LIN = VSL/VAP), oscillation of the track (WOB = VAP/VCL), and beat-cross frequency (BCF, Hz). The limit for distinguishing motile spermatozoa was set at VCL = 15 µm/s.

### 2.8. Statistics

Data on the content of hormones in blood plasma and testes in each experimental condition were checked for normality and homogeneity of dispersion using Kolmogorov–Smirnov and Levene’s tests, correspondingly. In groups with normally distributed and similar dispersion values, parametric one-way ANOVA was applied, and Tukey’s honestly significant difference test (HSD test) was used to find the significant differences between groups. In groups in which values were not normally distributed or had not similar dispersion values, the data were analyzed using Kruskal–Wallis test followed by the multiple comparisons of mean ranks for all groups. Results were presented as mean ± standard error (for motility percentage after TS IVM, blood plasma level of hormones and testis level of hormones). Relationships between hormone content and motility percentage after IVM were also explored by correlation and regression analysis. When the hypothesis that there is no relationship between hormone content and IVM was rejected (*p* < 0.05), the data were presented as a scatterplot, which additionally consists of information about the coefficient of determination and plotted linear regression lines.

Data on spermatozoon motility parameters of Wolffian duct sperm and in vitro matured testicular sperm were first explored for correlations using Spearman’s rank correlation coefficient (r_s_). To simplify the presentation of the data, only parameters with a low correlation coefficient (r_s_ < 0.12) were selected as descriptors of sperm motility. These parameters were VCL (curvilinear velocity) and LIN (linearity). Data set containing motility parameters from all video records (altogether from 219,379 spermatozoa) was subjected for processing to get average values for each combination of male/experimental group/post-activation time. These data were further averaged by males and were subjected to regression analysis of dependency of each parameter from post-activation time. Regression analysis was performed to find the best-fit of dependences by polynomial regression according to McDonald (2009) [16]. Finally, averaged data (as dots) and regression lines of polynomial regression of third-order are shown in figures.

All calculations and plotting were performed using Statistica (version 13, TIBCO software Inc., 2017, Palo Alto, CA, USA) and Microsoft Excel spreadsheet. Alpha was set at 0.05.

## 3. Results and Discussion

### 3.1. Effect of Environmental Temperature and Hormonal Stimulation on Sperm Motility Parameters

Sterlet testicular spermatozoa upon dilution in the activating solution did not become motile in any study groups. The acquisition of motility resulted only after incubation in seminal fluid from WS. The relative activity was not uniform for fish in various groups—in group OW, TS motility percentage was very low (7 ± 1%) (Figure 2), and it was not significantly changed (12 ± 2%) after fish adaptation to spawning temperature (group ST). However, it was significantly increased (up to 40 ± 5%) when fish adapted to spawning temperature were injected with CPP (group ST-HI).

Kinetic data on spermatozoa motility were obtained for many spermatozoa (219,379). Lines of polynomial regression obtained as a result of regression analysis were characterized by high r^2^. These facts give grounds for considering the presented regression lines for VCL and LIN (Figure 3) as general trends in changes of these parameters during the motility period. Accordingly, VCL of naturally matured Wolffian duct sperm (control condition) at 10 s post activation was around 125 µm/s (Figure 3A), which was lower than values found in our previous studies on sterlet [3,4], but similar to values in the study of Sieczynski et al. [17]. However, it should be noted here that CASA results can be highly influenced by different technical parameters of video microscopy (e.g., frame rate, magnification and type of chamber) [18]. This is why comparing the absolute values of spermatozoa kinematic parameters presented in different studies (and thus obtained by different equipment) cannot be considered a qualitative approach. For the present study, the general trends in VCL changes seem more purposeful for elucidation of the effect of temperature and hormonal stimulation on testicular spermatozoa motility after IVM.

Testicular spermatozoa after IVM were characterized by lower VCL compared with the value for WS (Figure 3A). Quite similar values of TS velocity after overwintering and fish adaptation to spawning temperature were found (in the range of 75–82 µm/s at 10 s post activation for groups OW and ST), while VCL was increased up to 102 µm/s after hormonal stimulation of spermiation (group ST-HI). No differences in LIN between all studied sperm samples were detected (Figure 3B).

In general, it has been reported that stimulation of spermiation by exogenous hormones has no significant influence on sperm quality parameters (sperm motility percentage, motility duration or spermatozoa velocity) (reviewed in [19]). However, different results can be found in the literature depending on the hormone used and fish species [20,21,22,23]. Less information in this regard is available for Acipenseriformes. Alavi et al. [24] have shown significantly different effects of used hormonal treatments (carp pituitary extract, Ovopel and two doses of gonadotropin-releasing hormone analog implants) on sterlet sperm motility percentage and velocity. At the same time, no difference in the effect of different hormonal treatments (CPP and different doses of luteinizing hormone-releasing hormone analog) on the percentage of motile spermatozoa over the entire period of the experiment (up to 4.5 days after injection) was shown in paddlefish *Polyodon spathula* by Linhart et al. [25]. Unfortunately, no comparisons of sperm motility parameters between hormonally treated and control fish were present in both studies [24,25].

It should be stressed here that all the above-discussed information concerns already matured sperm (collected from urogenital ducts). At the same time, in the current research, we present the effect of hormonal stimulation on the ability of sterlet testicular (immature) sperm to become matured (able to activate motility) in vitro. From the data obtained, we can conclude that the ability of sterlet testicular spermatozoa for final maturation is related to hormonal stimulation of spermiation, which leads to increased motility percentage and velocity after in vitro maturation, without changes in the linearity of spermatozoa track. It allows for the general suggestion that as a result of hormonal treatment, the physiological state of testicular spermatozoa is changed, resulting in an increased number of testicular spermatozoa able to fulfill final maturation during their transit through the kidney.

### 3.2. Effect of Environmental Temperature and Hormonal Stimulation on Content of Sex Steroid Hormones in Blood Plasma and Testes

#### 3.2.1. Hormone Content in Blood Plasma and Testes

Transfer of fish after overwintering from fish-farming ponds (water temperature 2 °C) into the 0.8 m^3^ tanks with the subsequent rise in water temperature to 14 °C did not result in increased production of sperm. Sperm samples were collected only from some males (ca 40%) in groups OW (1.9 ± 0.3 mL) and ST (2.0 ± 0.2 mL). The low number of spermiating males in these groups and the low volume of sperm produced are not surprising, as problems with spontaneous spermiation or low volume of milt in captivity are well-known for different fish species ([26,27]; reviewed in [19]). Additionally, it should be remarked that in the current study, experiments were done in advance of the natural spawning season. In artificial reproduction, these problems are easily overcome by carrying hormonal stimulation. Moreover, indeed, induction of spermiation by injection of CPP (group ST-HI) led to a collection of sperm from all males, and the volume of sperm collected was approximately five times bigger (11.3 ± 1.3 mL) than in the OW and ST groups.

The highest levels of all sex steroid hormones in sterlet blood plasma were detected in fish from group ST-HI. At the same time, the patterns of change for each hormone were slightly different. Plasma level of testosterone (T) was not different in groups OW and ST due to sterlet adaptation to spawning temperature (Figure 4A), while it significantly increased as a result of hormonal induction of spermiation in group ST-HI in comparison with group ST. The 11-ketotestosterone (11-KT) concentration in blood plasma started to rise (*p* = 0.097) due to fish adaptation to spawning temperature (Figure 4B) and reached a significantly higher value in group ST-HI (hormonal stimulation) compared with group OW. Hormonal induction of spermiation led to a significant increase in the concentration of 17,20β-dihydroxypren-4-en-3-one (17,20β-P) in group ST-HI than in groups OW and ST (Figure 4C). No change in blood plasma level of 17,20β-P was found after fish adaptation to spawning temperature.

Blood plasma levels of the measured androgens are known to be related to the spermatogenetic cycle in many fish species ([28,29,30], reviewed in [31]). In sturgeon, based on studies of T and 11-KT concentrations during gonadal development in giant sturgeon *Huso huso*, Russian sturgeon *A. gueldenstaedtii* and stellate sturgeon *A. stellatus* [32], Chinese sturgeon *A. sinensis* [33], white sturgeon *A. transmontanus* [34], shovelnose sturgeon *Scaphirhynchus platorynchus* [35], sterlet [36], bester (F1 sturgeon hybrid *H. huso* female x *A. ruthenus* male) [10], and Siberian sturgeon *A. baerii* [37], it can be summarized that blood plasma content of sex steroids increases with the progress in gonad preparation towards spawning. This is why the levels of these hormones in fish of groups OW and ST observed in the present study can be considered corresponding to the pre-spermiation stage, when all seminiferous tubules are already full of spermatozoa that are ready to be released after hormonal stimulation. Confirmation of this supposition will be presented below (see Section 3.3). Further, hormonal stimulation results in quite variable changes of sex steroid levels in the blood of different sturgeon species (sterlet [24], Siberian sturgeon [37], giant sturgeon and Russian sturgeon [32] and stellate sturgeon [32,38]) and other fish species (e.g., striped bass *Morone saxatilis* [39] and meagre *Argyrosomus regius* [23]). This wide spectrum of responses of sex steroids does not allow for generalization and cannot be unambiguously explained. For all that, concerning the fact that in the current study, blood plasma levels of both androgens (T and 11-KT) and 17,20β-P were significantly increased after CPP injection (Figure 4), we consider this additional rise in sex steroid contents as a descriptor of the physiological state of the fish under study.

Since sex steroid hormones are synthesized in gonads, their blood plasma concentration can differ from their gonadal level. There is an opinion that measurements of circulating levels of these hormones “should be completed by an analysis of the gonadal tissues where the target cells are close to the secreting ones” [40] (p. 279). Notwithstanding the above, information is extremely scarce on the gonadal level of sex steroid hormones in fish. In the present study, the gonadal level of sex steroid hormones (Figure 5) was significantly different between groups only for 11-KT. Individual variability of hormone concentrations in all studied fish groups was so high that it did not allow any conclusions related to the effects of adaptation to spawning temperature and hormonal induction of spermiation on their level. The 11-KT concentration was significantly higher in group ST-HI (induction of spermiation) than in groups OW and ST (Figure 5B).

Taking into account that the available published data mainly concern in vitro steroidogenesis by gonads [41,42,43,44] and the suggestion of Bukovskaya et al. [42] about the extra-gonadal origin of 11-oxygenated androgens in Russian sturgeon, further studies of the gonadal level of sex steroid hormones in different sturgeon species are strongly needed.

#### 3.2.2. Hormone Content in Relation to Testicular Sperm IVM

Performed correlation analysis allowed to find significant dependences of motility percentage of sterlet testicular sperm after IVM on contents of 17,20β-P in blood plasma and 11-KT in gonads (Figure 6). Similar correlations were also found between contents of 17,20β-P in blood plasma and 11-KT in gonads and VCL of testicular spermatozoa at the initial stage of motility after IVM (for 17,20β-P and 11-KT r^2^ was 0.2536 and 0.1949, respectively). Relationships between blood T and 11-KT contents, gonad T and 17,20β-P contents and motility percentage and VCL of testicular sperm after IVM were not statistically significant.

In teleosts, 11-KT is considered to be the major androgen involved in regulating different stages of spermatogenesis and initiation of spermiation, while 17,20β-P, as well as 17,20,21-trihydroxy-4-pregnen-3-one, mainly function as maturation-inducing steroids ([45,46]; reviewed in [19,31,47,48,49,50]). Final spermatozoa maturation in sperm duct (not in testes) due to the presence of different factors (pH, potassium and bicarbonate ions) is known for salmonids and Japanese eel *Anguilla japonica* [51,52], and it has been suggested that the direct action of 17,20β-P on spermatozoon membrane is needed for this process [47,50]. Less information related to the mentioned steroids is available for sturgeon [10,32,38,53]. The significant dependence of sterlet testicular sperm motility percentage after IVM on 17,20β-P content in blood plasma (Figure 6A) may argue for the possibility that a significant increase in blood plasma 17,20β-P content along with a rise in gonadal 11-KT concentration after hormonal induction of spermiation in group ST-HI can stimulate certain intracellular signaling pathways and, in such a way, prepare TS for final maturation upon contact with urine. The maturational effect of 17,20β-P can be realized, e.g., through the activation of carbonic anhydrase, like it was suggested for the Japanese eel [47,50]. For revealing the exact mechanisms of the effects of sex steroids on sperm maturation in sturgeon, identifying androgen and/or progestogen receptors on the sturgeon spermatozoon membrane is highly required. While the necessity of the study in this direction has already been stressed [6], information on the topic is not still available. Additionally, based on the facts that in sturgeons and paddlefishes: (1) spermatozoa motility is under the control of potassium and calcium ions, pH and osmolality [54,55,56], (2) sperm sensitivity to calcium ions can fluctuate during the maturation process [57] and (3) hormonal treatment can influence ion levels and osmolality of seminal fluid [58], the participation of ions, pH and osmolality in sperm maturation process in Acipenseriformes cannot be excluded and needs further study.

### 3.3. Testes Histological Identification

For sterlet males used in the experiment, histological identification of gonads did not reveal any crucial differences resulting from fish adaptation to spawning temperature and hormonal stimulation of spermiation. In testes of fish from all studied groups, practically all seminiferous tubules were full of spermatozoa (Figure 7, mid-region). Herewith, it should be noted that in testes of fish from any group, areas containing spermatocysts with germ cells at earlier development stages (spermatogonia and spermatocytes), but probably not with spermatids were found (Figure 7, lateral side). It corresponds to study of Amiri et al. [10], who have found a more advanced stage of spermatogenic development in the vicinity of the efferent duct than at the lateral side of the testis in the bester. Our data about the presence of spermatocysts with germ cells at earlier development stages are also in line with the observation of Flynn and Benfey [59] about the presence of spermatozoa and spermatogonia, but not noticed spermatids in testes of spermiating shortnose sturgeon *A. brevirostrum*.

According to the definitions of Amiri et al. [10], sterlet males from all studied groups were at the pre-spermiation stage. As documented by Amiri et al. [10], at this stage in the bester large amounts of spermatozoa were released from spermatocysts starting from November and were maintained throughout the winter, while no sign of spontaneous spermiation and no sperm release after hand stripping were observed. A similar situation, when seminiferous tubules were full of spermatozoa, was also described for the testes of several sturgeon species [35,60,61,62,63,64,65,66,67,68]. All these demonstrate that release of spermatozoa from seminiferous tubules in sturgeon can occur starting at the early overwintering period, irrespectively of the location of the species’ natural habitat or farming conditions. Our data about the possibility of sperm collection even in some fish from groups OW (after overwintering) and ST (adapted to spawning temperature) are in line with the information presented above. At the same time, the found effect of hormonal treatment on the ability of testicular spermatozoa to fulfill final maturation in in vitro conditions makes our study important for understanding the potential of using in vitro matured testicular spermatozoa in aquaculture or conservation programs, which can be realized in cases of accidental death of valuable broodstock or failure to obtain Wolffian duct sperm of high-quality.

## 4. Conclusions

The ability of sterlet testicular sperm obtained in advance of the spawning season to mature in vitro depends on the hormonal treatment required for artificial sturgeon propagation in farmed conditions. Off-season adaptation of sterlet to spawning temperatures did not increase the volume and motility percentage of Wolffian duct sperm or the motility percentage of testicular sperm after in vitro maturation. Hormonal stimulation of spermiation in fish adapted to spawning temperature has led to increases in volume and motility percentage of Wolffian duct sperm and a significant rise of motility percentage and velocity of testicular sperm after in vitro maturation. These increases were associated with elevation of circulating levels of T, 11-KT and 17,20β-P, but a rise of only 11-KT level in testes. Histological analysis revealed that fish from all studied groups were at the same testicular developmental stage, pre-spermiation. Taken together, the data demonstrate that the ability for testicular spermatozoa to be matured in vitro can be achieved after male overwintering, while hormonal stimulation of spermiation is an absolute requirement for optimal in vitro maturation. Quite probably, hormonal stimulation results not only in the production of seminal fluid, but also in forming specific intratesticular conditions at the final stage of spermiogenesis, which allows the preparation of testicular spermatozoa for further final maturation upon contact with urine.

## Figures and Tables

**Figure 1 animals-11-01417-f001:**
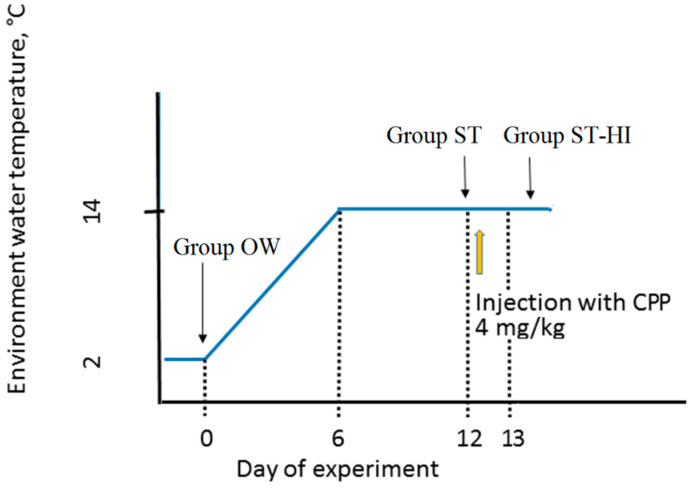
Temperature regime and hormonal injection timing for experimental fish groups. Group OW—fish after overwintering; group ST—fish adapted to spawning temperature; and group ST-HI—fish adapted to spawning temperature with hormonal induction of spermiation by carp pituitary powder (CPP).

**Figure 2 animals-11-01417-f002:**
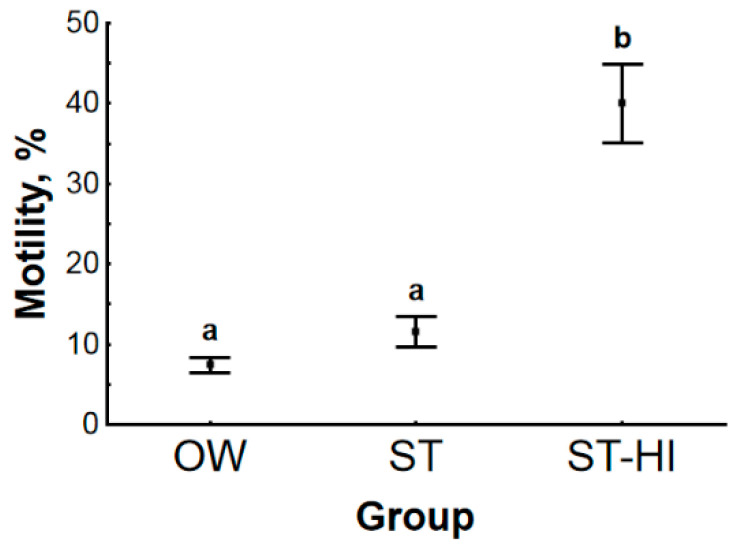
Motility percentage after testicular sperm in vitro maturation in experimental groups of sterlet. OW—fish were held in fish-farming ponds (water temperature 2 °C); ST—fish were adapted to spawning temperature of 14 °C, spermiation was not induced; ST-HI—fish were adapted to spawning temperature of 14 °C, spermiation was induced. Dots represent the means, and whiskers represent the standard errors. Values with different letters are significantly different (*p* < 0.05, multiple comparisons of mean ranks for all groups).

**Figure 3 animals-11-01417-f003:**
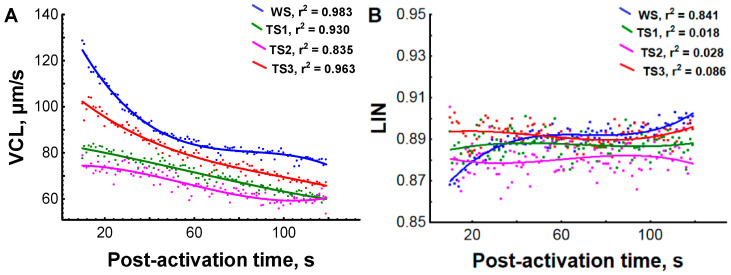
Sperm motility parameters of Wolffian duct sperm and in vitro matured testicular sperm from experimental groups of sterlet at different post-activation times. (**A**) Curvilinear velocity (VCL, µm/s), (**B**) linearity (LIN). TS1—testicular sperm from fish of group OW, held in fish-farming ponds (water temperature 2 °C); TS2—testicular sperm from fish of group ST, adapted to spawning temperature of 14 °C, spermiation was not induced; TS3—testicular sperm from fish of group ST-HI, adapted to spawning temperature of 14 °C, spermiation was induced; WS—sperm collected from Wolffian duct (naturally matured) from fish of group ST-HI. Each dot represents an average value for each combination of male/experimental group/post-activation time. Lines of polynomial regression of third-order and corresponding r^2^ are presented. Different colors of dots and lines correspond to the used sperm sample: blue—from WS, green—from TS1, purple—from TS2, red—from TS3.

**Figure 4 animals-11-01417-f004:**
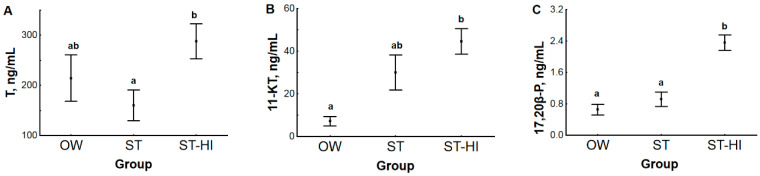
Blood plasma level of: (**A**) testosterone (T, ng/mL), (**B**) 11-ketotestosterone (11-KT, ng/mL), (**C**) 17,20β-dihydroxypren-4-en-3-one (17,20β-P, ng/mL) in experimental groups of sterlet. OW—fish were held in fish-farming ponds (water temperature 2 °C); ST—fish were adapted to spawning temperature of 14 °C, spermiation was not induced; ST-HI—fish were adapted to spawning temperature of 14 °C, spermiation was induced. Dots represent the means, and whiskers represent the standard errors. Values with different letters are significantly different (for T and 17,20β-P, *p* < 0.05, Tukey’s HSD test, and for 11-KT *p* < 0.05, multiple comparisons of mean ranks for all groups).

**Figure 5 animals-11-01417-f005:**
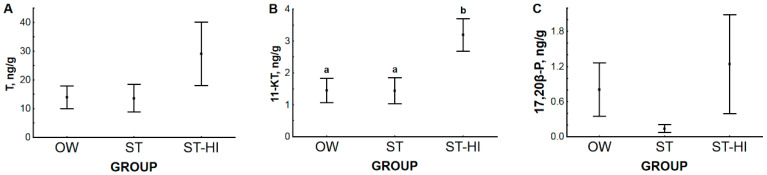
Testis level of: (**A**) testosterone (T, ng/g), (**B**) 11-ketotestosterone (11-KT, ng/g), (**C**) 17,20β-dihydroxypren-4-en-3-one (17,20β-P, ng/g) in experimental groups of sterlet. OW—fish were held in fish-farming ponds (water temperature 2 °C); ST—fish were adapted to spawning temperature of 14 °C, spermiation was not induced; ST-HI—fish were adapted to spawning temperature of 14 °C, spermiation was induced. Dots represent the means, and whiskers represent the standard errors. Values with different letters are significantly different (*p* < 0.05, multiple comparisons of mean ranks for all groups).

**Figure 6 animals-11-01417-f006:**
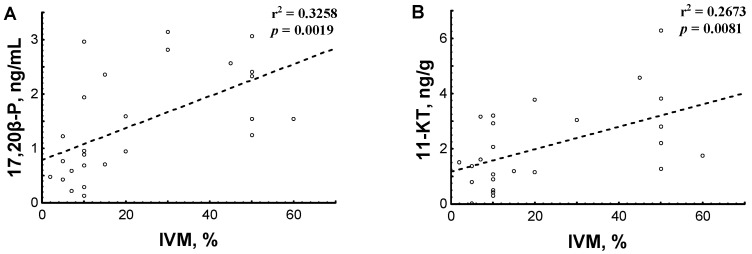
Relationship between sterlet testicular sperm motility percentage after in vitro maturation (IVM) and hormone content in: (**A**) blood plasma, 17,20β-dihydroxypren-4-en-3-one (17,20β-P, ng/mL), and (**B**) gonad, 11-ketotestosterone (11-KT, ng/g gonad). Scatterplots, linear regression lines, coefficient of determination r^2^ and *p* for checking the hypothesis that there is no relationship between hormone content and testicular sperm motility percentage after IVM are presented. The data obtained from all experimental sterlet males are presented.

**Figure 7 animals-11-01417-f007:**
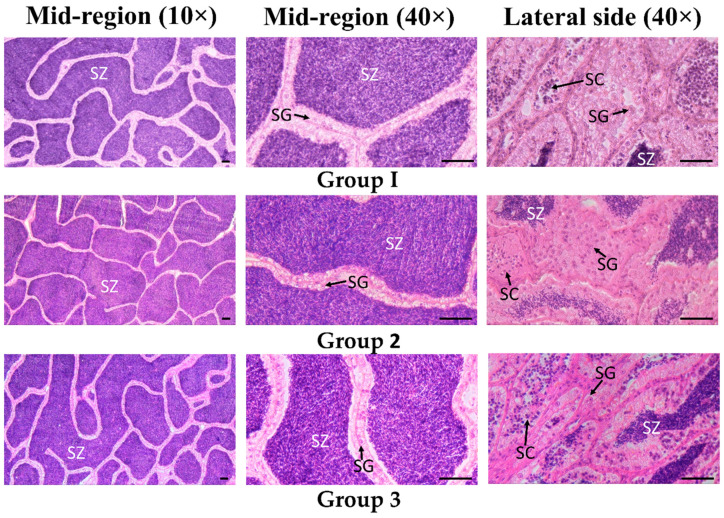
Light micrographs of testis sections in different fish groups. Group OW—fish were held in fish-farming ponds (water temperature 2 °C); group ST—fish were adapted to spawning temperature of 14 °C, spermiation was not induced; group ST-HI—fish were adapted to spawning temperature of 14 °C, spermiation was induced. SC—spermatocytes; SG—spermatogonia; SZ—spermatozoa. Mid-region (10×)—predominant images of the mid-regions of the testis, magnification 10×; mid-region (40×)—predominant images of the mid-regions of the testis, magnification 40×; lateral side (40×)—images of the lateral side of the testis, magnification 40×. Scale = 50 µm.

## Data Availability

The data presented in this study are available on request from the corresponding author.

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
