# Peer review of "Influence of Environmental Temperature and Hormonal Stimulation on the In Vitro Sperm Maturation in Sterlet Acipenser ruthenus in Advance of the Spawning Season"

_animals, 2021, doi:10.3390/ani11051417_

Round 1

Reviewer 1 Report

The title: In the title the “in advance to the spawning season” should be added.

Lines 41-43: Where the level of the hormones increased? In blood plasma?

Lines 79-80: What do you mean by “specific intratesticular conditions”? What are the conditions which “prepare testicular spermatozoa”?

Lines 99-105: Was it their spawning time (season)? It seems to me that it was in advance to the proper spawning season. The reader need to be aware of that fact.

Lines 173; 174-175: When the motility rate was estimated? Just after collection of sperm from WS?

Lines 178-180: Please, rephrase this sentence.

Line 184: WS fluid obtained from only one male was taken as a maturation fluid? This part is not clear and it is very important to understand that only single WS fluid was used as a maturing agent.

Lines 185-187: What do you mean here? Do you mean that you chose WS fluid which gave the best results of incubation in some preliminary experiment? This need to be clarified.

Line 301: I suggest to discuss also whether the fish used were taken during the spawning season or it was I advance to the spawning season. Many fish species taken prior the spawning season from the water temperature of 2°C and acclimated so fastly would have immature sperm what could require hormonal support to trigger proper spermiation. Please, also consider in the Discussion how this could affect the results observed and what could have been observed during the spawning season.

Lines 302-3-4: Please, explain what you are considering as mature and what as immature here (mature – striped via the duct?).

Lines 308-311: Do you mean, that hormonal stimulation simply triggered maturation processes in sperm? If yes, please, state that clearly. This sentence is unclear.

Line 318: From how many males sperm was obtained? Anyway, this information is missing for all the groups.

Lines 319-321: And maybe the experiment was performed a way to early (prior the spawning season)? Please, refer to that.

Lines 349-400, 411-414: I suggest to remove this part. The Authors are barely discussing here any of their results but providing kind of review on how the hormones works in sturgeon males. There is also kind of description on why the Authors observed different levels of steroids in gonads and blood plasma. Actually, these three paragraphs can be easily compressed into single elegant paragraph where direct link with the study aim and the results obtained should be the priority. This would also enable to make significant shortage of the article – with huge benefit to it. This is straightforward study and readers will appreciate its straightforward form.

Lines 416-421: And what about correlation with other sperm motility indices (VCL, for instance)?

Lines 435-480: This need to be shortened by half, at least! I got completely lost what the Authors have in mind. Please, keep the line with the aim of the study. For me it is obvious that the sperm in OW and ST groups was simply immature due to lack of the steroidegenic effect triggering their final maturation. The Authors are going deeply into the mechanisms unnecessarily – this makes this MS hardly readable and very long.

Lines 483-495; 505-533: Also in this part there is barely link to the study results. The message is clear: only sperm after the hormonal stimulation have matyred and were released easily from the globes (this is obvious when taking a look at the results where only few fish were spermiating in OW and ST groups). Please, make the message clear and stick to the aim of the study, avoid excessive speculations.

Lines 505-515: This paragraph should be the main actor in this paragraph. It is now clear that the study was performed too early (in advance of the spawning season). So, please, refer to the fact how this study is specific to the advanced spawning of sterlet.

Lines 516-533: To be removed.

Lines 535-536: This sentence should be changed to: “The ability of sterlet testicular sperm obtained in advance to the spawning season to mature in vitro depends on hormonal treatment.”

Line 357: Add at the beginning of this sentence “Off season adaptation…”

Figure 3a: Where the differences between groups statistically significant?

Author Response

We greatly appreciate the thorough evaluation of our manuscript and your comments. We hope that our corrections improved the text. Our answers follow below. In our answers, line numbers correspond to version “animals-1179120-Revised”.

Comments

The title: In the title the “in advance to the spawning season” should be added.

Viktoriya Dzyuba’s answer (VD): “in advance to the spawning season” was added. Please, see Line 4.

Lines 41-43 (now Lines 40–42): Where the level of the hormones increased? In blood plasma?

VD: Concentrations of all three studied hormones were significantly increased in blood plasma. Text was corrected. Please, see Line 40.

Lines 79-80: What do you mean by “specific intratesticular conditions”? What are the conditions which “prepare testicular spermatozoa”?

VD: We mean here that hormonal treatment can not only provoke production of seminal fluid, but also can result in certain changes in testis (pH, concentration of ions, osmolality, stimulation of signaling pathways with involvement of putative receptors for sex steroids on spermatozoon membrane). All these changes can supposedly “prepare testicular spermatozoa” for final maturation upon transit through kidneys. We did not want to complicate and artificially increased Introduction part, but all these points were mentioned in discussion (please, see Lines 507–552). Now they are still mentioned (Lines 482–506), but, please, remark that this part was shortened according to your suggestions to Lines 435–480.

Lines 99-105: Was it their spawning time (season)? It seems to me that it was in advance to the proper spawning season. The reader need to be aware of that fact.

VD: For sturgeon, precise time of natural spawning is not easy to be defined, as it strictly depends on water temperature. For example, for sterlet it starts from 7.5–10.0 °C (Chebanov and Galich, 2013, FAO, Fisheries and Aquaculture Technical Paper N 558). In aquaculture of the Czech Republic sturgeon reproduction can be performed starting from the middle of winter, but usually it is performed at the beginning of spring. Taking the latter in mind, you are right that our study was done in advance to the proper spawning season. We hope that now, after addition the information that it was in advance to the spawning season in the title and in the text (Lines 331, 625, 627), it is clearer for readers.

Lines 173; 174-175 (now Lines 176–178): When the motility rate was estimated? Just after collection of sperm from WS?

VD: Motility rate was evaluated within half an hour after collection of sperm from Wolffian duct. It was added to the text. Please, see Lines 178–179.

Lines 178-180 (now Lines 182–184): Please, rephrase this sentence.

VD: It was rephrased from “Supernatants from Wolffian duct sperm were designated as seminal fluids from Wolffian duct sperm and used for IVM of TS” to “Supernatants obtained after the last centrifugation were used for IVM of TS”. Please, see Lines 183–184.

Line 184 (now Line 189): WS fluid obtained from only one male was taken as a maturation fluid? This part is not clear and it is very important to understand that only single WS fluid was used as a maturing agent.

VD: Seminal fluid of WS from only one male was used for several reasons: it was of the highest volume (Please, see Line 188), and its maturational effect did not differ from other fluids. The last fact was mentioned below (Please, see Lines 190–192). To make this message clearer we slightly corrected the text (Lines 187, 189).

Lines 185-187 (now Lines 189–192): What do you mean here? Do you mean that you chose WS fluid which gave the best results of incubation in some preliminary experiment? This need to be clarified.

VD: We mean that maturational effect of the selected seminal fluid was not different from other seminal fluids. We just selected one of such fluid (of the highest volume and the same maturational effect) to standardize the experimental conditions. Corrections were made in text (Lines 187, 189).

Line 301 (now Line 308): I suggest to discuss also whether the fish used were taken during the spawning season or it was I advance to the spawning season. Many fish species taken prior the spawning season from the water temperature of 2°C and acclimated so fastly would have immature sperm what could require hormonal support to trigger proper spermiation. Please, also consider in the Discussion how this could affect the results observed and what could have been observed during the spawning season.

VD: The fact that the experiment was conducted in advance to spawning season is now mentioned in the title and in the text (Lines 331, 625, 627). At the same time, we do not think that temperature increase within six days with subsequent six day holding fish at the reached temperature is too fast. According to Sturgeon Hatchery Manual (Chebanov and Galich, 2013, FAO, Fisheries and Aquaculture Technical Paper N 558), sturgeon transition out of the overwintering should be done with a temperature gradient 2–3 oC per day with a period of holding at constant temperature. In our study it was 2 oC per day followed by six days at constant temperature.

Lines 302-3-4 (now Lines 309–312): Please, explain what you are considering as mature and what as immature here (mature – striped via the duct?).

VD: In sturgeon, we consider sperm striped via urogenital duct as mature one. Testicular sperm (which cannot activate motility in activating medium) is immature one. These definitions are based on our previous findings that final sperm maturation (as ability to activate motility and fertilize) in sturgeon occurs after sperm transit through the kidney, where sperm is mixed with urine. To clarify this, the text was changed. Please, see Line 310.

Lines 308-311 (now Lines 316–319): Do you mean, that hormonal stimulation simply triggered maturation processes in sperm? If yes, please, state that clearly. This sentence is unclear.

VD: As maturation of testicular spermatozoa takes place during their transit through the kidney,  we cannot say unambiguously that hormonal stimulation simply triggered this process. We can only say that in fish which were injected with CPP there was an increase in number of testicular spermatozoa which are able to become matured in vitro. We hope that after correction made according to your previous comment there is no need to make additional corrections.

Line 318 (now Line 326): From how many males sperm was obtained? Anyway, this information is missing for all the groups.

VD: Sperm was obtained from around 40% of males in groups OW, ST and from all males in group ST-HI. This information was added to the text. Please, see Lines 327, 333–335.

Lines 319-321 (now Lines 327–329): And maybe the experiment was performed a way to early (prior the spawning season)? Please, refer to that.

VD: The fact that experiment was performed in advance to the spawning season in now mentioned in the text. Please, see Lines 330–331.

Lines 349-400, 411-414 (now Lines 381–399, 412–444, 455–458): I suggest to remove this part. The Authors are barely discussing here any of their results but providing kind of review on how the hormones works in sturgeon males. There is also kind of description on why the Authors observed different levels of steroids in gonads and blood plasma. Actually, these three paragraphs can be easily compressed into single elegant paragraph where direct link with the study aim and the results obtained should be the priority. This would also enable to make significant shortage of the article – with huge benefit to it. This is straightforward study and readers will appreciate its straightforward form.

VD: We agree that this part of the text can be shortened. It was done, please, see Lines 360–380, 400–411. At the same time, as presented information is related to the data for blood and testis, we prefer to put it into two different paragraphs. So, three mentioned paragraphs were shortened and compressed into two ones.

Lines 416-421 (now Lines 460–468): And what about correlation with other sperm motility indices (VCL, for instance)?

VD: Similar correlations were also found between contents of 17,20β-P in blood plasma and 11-KT in gonads and VCL of testicular spermatozoa at the initial stage of motility after IVM (for 17,20β-P and 11-KT r2 was 0.2536 and 0.1949, respectively). This information was added to the text. Please, see Lines 463–466.

Lines 435-480 (now Lines 507–552): This need to be shortened by half, at least! I got completely lost what the Authors have in mind. Please, keep the line with the aim of the study. For me it is obvious that the sperm in OW and ST groups was simply immature due to lack of the steroidegenic effect triggering their final maturation. The Authors are going deeply into the mechanisms unnecessarily – this makes this MS hardly readable and very long.

VD: This part of the text was shortened approximately in half. Please, see Lines 482–506. At the same time, we would like to mention here that all fish in our experiment were mature. Sterlet testicular spermatozoa are immature, and they are immature in all studied groups. It is related to specific structure of sturgeon reproductive system and specific for sturgeon sperm maturation process. Hormonal treatment in group ST-HI increased the number of TS which can be matured in vitro.

Lines 483-495; 505-533 (now Lines 555–567; 595–623): Also in this part there is barely link to the study results. The message is clear: only sperm after the hormonal stimulation have matyred and were released easily from the globes (this is obvious when taking a look at the results where only few fish were spermiating in OW and ST groups). Please, make the message clear and stick to the aim of the study, avoid excessive speculations.

VD: Related to your comment to Lines 483–495 (now Lines 555–567), we would like to keep this part of the text unchanged, as exactly here we describe our results (Lines 555–561) and shortly discuss them in two next sentences (Lines 561–567). Answers to your comment to Lines 505–533 (now Lines 595–623) are presented below.

Lines 505-515 (now Lines 595–605): This paragraph should be the main actor in this paragraph. It is now clear that the study was performed too early (in advance of the spawning season). So, please, refer to the fact how this study is specific to the advanced spawning of sterlet.

VD: We agree with you that this paragraph is the main actor in discussion of our results on histology. To make this part of the text clearer for readers and taking into account your previous comments, the text was modified. Please, see Lines 578–594.

Lines 516-533 (now Lines 606–623): To be removed.

VD: Part of this paragraph was removed, and text was re-arranged. Please, see Lines 606–623.

Lines 535-536 (now Lines 625–627): This sentence should be changed to: “The ability of sterlet testicular sperm obtained in advance to the spawning season to mature in vitro depends on hormonal treatment.”

VD: This sentence was changed: “obtained in advance to the spawning season” was added. Please, see Line 625.

Line 357 (now Line 627): Add at the beginning of this sentence “Off season adaptation…”

VD: We think that you mean here Line 537 (now Line 627). “Off-season” was added at the beginning of the sentence. Please, see Line 627.

Figure 3a: Where the differences between groups statistically significant?

VD: We used regression analysis to find the best fit lines for description of general trend of dynamics of VCL change (see Lines 237–240). This analysis does not suppose statistical comparisons of groups but taking into account big number of individually measured parameters involved into analysis (from  219,379 individual spermatozoa) and high r2 (this aspect are presented in the text of manuscript), we consider this kind of presentation as relevant for description of general trends in VCL and did not use other statistical methods for comparisons of groups.

Reviewer 2 Report

The submitted manuscript “Influence of environmental temperature and hormonal stimulation on the in vitro sperm maturation in sterlet Acipenser ruthenus” by Dzyuba et al. Examines whether sterlet surgeon testicular sperm complete final maturation at different stages of the male reproductive cycle via analysis of testicular sperm kinematic parameters after in vitro maturation. Overall, the abstract and introduction are very nicely written. I have no changes there.

Methods: Can any details be provided on overwintering conditions as likely they change from year to year.  I see it is briefly described in next paragraph, but likely more husbandry conditions are needed – diets, water quality, etc., etc.,

Line 129: What was dose MS-222 or clove oil.?

Great stats section. I especially liked the use of Macdonald 2009 for model fit – not many folks do that, but it is correct.

Line 237-238: Why not say “Alpha was set at 0.05”.

Line 261-262: which was lower.

Fig3. B. I don’t see the point of this, as LIN is not usually corrected well to reproductive performance. If needed make scale between 0.80 and 0.95 or something to see trends, or lack there of.

Fig 6: It appears these figures could benefit from log trans on both axis. Likely get better function thereafter, did you try this?

Table 1 can be replaced with 1 sentence in text.

Fig 7: Instead of A B C on top why not just put the treatments, as you have the room.

Line 545: Please replace testify with something like demonstrate.

Nice work

Author Response

We greatly appreciate the thorough revision of our manuscript, your comments and positive evaluation of our study. We hope that our corrections improved the text. Our answers follow below. In our answers, line numbers correspond to version “animals-1179120-Revised”.

Comments

The submitted manuscript “Influence of environmental temperature and hormonal stimulation on the in vitro sperm maturation in sterlet Acipenser ruthenus” by Dzyuba et al. Examines whether sterlet surgeon testicular sperm complete final maturation at different stages of the male reproductive cycle via analysis of testicular sperm kinematic parameters after in vitro maturation. Overall, the abstract and introduction are very nicely written. I have no changes there.

Methods: Can any details be provided on overwintering conditions as likely they change from year to year.  I see it is briefly described in next paragraph, but likely more husbandry conditions are needed – diets, water quality, etc., etc.,

Viktoriya Dzyuba’s answer (VD): During overwintering period fish were not fed, pond water pH was 8.3–8.7, contents of NH4+ and NO3- were around 0 mg/mL, oxygen content was higher than 95% of saturation level. All this information was added to the text. Please, see Lines 107–108.

Line 129 (now Line 131): What was dose MS-222 or clove oil.?

VD: There was no anaesthesia, it was euthanasia. Fish were euthanized by striking the cranium followed by exsanguination. This information was added to the text. Please, see Line 132.

Great stats section. I especially liked the use of Macdonald 2009 for model fit – not many folks do that, but it is correct.

VD: Thank you very much.

Line 237-238 (now Lines 242–243): Why not say “Alpha was set at 0.05”.

VD: We agree, text was changed. Please, see Line 242.

Line 261-262 (now Lines 267–268): which was lower.

VD: It was corrected (“that” was replaced by “which”). Please, see Line 268.

Fig3. B. I don’t see the point of this, as LIN is not usually corrected well to reproductive performance. If needed make scale between 0.80 and 0.95 or something to see trends, or lack there of.

VD: We agree that correlation between LIN and reproductive performance is not easy to find. At the same time, it is a parameter which can be considered as a simple descriptor of sperm track shape. That is why we would like to keep it and have modified figure according to your suggestion. Please, see Line 283.

Fig 6: It appears these figures could benefit from log trans on both axis. Likely get better function thereafter, did you try this?

VD: Yes, we did try, and no benefits were found after transformations. E.g. after Log10/Log10 transformations, r2 = 0.3746 for 17,20β-P (a little bit higher then presented in the manuscript, without transformation), and for 11-KT r2 = 0.2014 (even a little bit lower then presented in the manuscript, without transformation). That is why we would like to keep this part of data presentation without changes.

Table 1 can be replaced with 1 sentence in text.

VD: Table 1 was deleted, and the text was changed. Please, see Lines 477–480, 466–468. The sentence related to the table in section 2.8 was also deleted (Lines 227–228).

Fig 7: Instead of A B C on top why not just put the treatments, as you have the room.

VD: Thank you for the suggestion. Treatments were put on the top of figure, figure legend was modified. Please, see Lines 569, 574–576.

Line 545 (now Line 636): Please replace testify with something like demonstrate.

VD: “Testify” was replaced by “demonstrate”. Please, see Line 636.

Nice work

VD: Thank you very much for your interesting comments and kind evaluation of our work.